

# Adaptation and potential culture of wild Amphipods and Mysids as potential live feed in aquaculture: a review

Hidayu Suhaimi[1], Muhammad Irfan Abdul Rahman[1], Aisyah Ashaari[1], Mhd Ikhwanuddin[2] and Nadiah Wan Rasdi[1]

[1] Faculty of Fisheries and Food Science, Universiti Malaysia Terengganu, Kuala Nerus, Terengganu, Malaysia
[2] Higher Institute Centre of Excellence (HICoE), Institute of Tropical Aquaculture and Fisheries, Universiti Malaysia Terengganu, Kuala Nerus, Terengganu, Malaysia

Corresponding author
Nadiah Wan Rasdi,
nadiah.rasdi@umt.edu.my

## ABSTRACT

Live foods such as phytoplankton and zooplankton are essential food sources in aquaculture. Due to their small size, they are suitable for newly hatched larvae. *Artemia* and rotifer are commonly used live feeds in aquaculture; each feed has a limited dietary value, which is unsuitable for all cultured species. Whereas, copepod and cladocerans species exhibit favorable characteristics that make them viable candidates as sources of essential nutrients for hatchery operations. Due to their jerking movements, it stimulates the feeding response of fish larvae, and their various sizes make them suitable for any fish and crustacean. Even though *Artemia* is the best live feed due to its proficient nutritional quality, the cost is very expensive, which is about half of the production cost. A recent study suggests the use of amphipods and mysids as alternative live feeds in aquaculture. High nutritional value is present in amphipods and mysids, especially proteins, lipids, and essential fatty acids that are required by fish larvae during early development. Amphipods and mysids are considered abundant in the aquatic ecosystem and have been used by researchers in water toxicity studies. However, the culture of amphipods and mysids has been poorly studied. There is only a small-scale culture under laboratory conditions for scientific research that has been performed. Thus, further research is required to find a way to improve the mass culture of amphipods and mysids that can benefit the aquaculture industry. This review article is intended to provide the available information on amphipods and mysids, including reproductive biology, culture method, nutritional value, feed enhancement, and the importance of them as potential live feed in aquaculture. This article is useful as a guideline for researchers, hatchery operators, and farmers.

## INTRODUCTION

In the aquaculture industry, live food is typically used as a feed for larvae or fingerlings of fish and crustaceans, with an emphasis on essential amino acids and fatty acids, as well as

nutrients, vitamins, and minerals, to provide the inadequate amount of proteins and lipids needed for larvae growth (*Hamre et al., 2013*). Live feeds are planktonic, observable by naked eyes, and have a peculiar motion that draws larvae and crustaceans to eat and capture them (*Samir & Banik, 2015*). Caprellid amphipods, commonly known as skeleton shrimps, are small marine crustaceans that are found in plenty in many littoral habitats (*Ros & Guerra-García, 2012*). Here they form an important trophic link between primary producers and higher trophic levels (*Woods, 2009*; *Ros & Guerra-García, 2012*). Live feed such as amphipod has the potential to be offered as a substitute feed for aquaculture, whether it is still alive or dead (*Baeza-Rojano, Hachero-Cruzado & Guerra-García, 2014*). They are a dominant species of the benthic fauna and often have high diversity (*Cunha, Moreira & Sorbe, 2000a*). Amphipod exhibits significant morphological adaptations that enable it to inhabit a wide range of environmental conditions (*Baeza-Rojano, Hachero-Cruzado & Guerra-García, 2014*). Amphipods have been shown to be important components of aquatic food webs in the wild due to their ability to transport nutrients and energy to higher trophic levels. As a result, it is ecologically acceptable to employ them as aquaculture species in captive and experimental settings. Amphipod feeds might be used in aquaculture to replace *Artemia* in different nursery feeding regimens, as a fishmeal substitute in aquafeeds, or even as a component of IMTA systems (*Shahin et al., 2023a*). Mysids are small shrimp-like crustaceans, a natural motile epibenthic invertebrate that is widely spread across marine environments, particularly in brackish, estuarine, coastal, and oceanic areas (*Verslycke et al., 2004*; *Oliveira et al., 2023*). Mysids are omnivorous and cannibal species, which feed on the diatom and another small crustacean (*Berezina, Razinkovas-Baziukas & Tiunov, 2017*). They often prey on small aquatic species, such as phytoplankton, zooplankton, and detritus (*O'Malley & Bunnell, 2014*). Mysids have been commonly used in water toxicity studies as an indicator (*Punchihewa, Krishnarajah & Vinobaba, 2017*).

Live feed is particularly essential for the growth of larval forms in aquaculture, as it is more easily ingested and digested (*Kandathil Radhakrishnan et al., 2020*), does not affect the water quality, and contains essential growth factors such as fatty acids and amino acids (*Turcihan et al., 2022*). According to *Kandathil Radhakrishnan et al. (2020)*, common live feeds consist of copepods, freshwater cladocerans (like *Daphnia* sp. and *Moina* sp.), and rotifers (like *Brachionus* sp.). These organisms are known for their high rate of reproduction, capacity to grow rapidly, and resilience to harsh environments. *Daphnia* sp. and *Moina* sp. are freshwater cladocerans that play an important role as live prey for freshwater fish culture (*Turcihan et al., 2022*). Beyond that, due to their high levels of protein and mineral contents, *Daphnia* sp. and *Moina* sp. are desirable alternative protein sources for replacing fish meal in fish diets (*Abo-Taleb et al., 2020*, *2021*; *Ashour et al., 2021*; *Turcihan et al., 2022*; *Suhaimi et al., 2022a*). While marine aquaculture is mainly dependent on limited live foods such as *Artemia*, rotifers, and copepods as feed for the larvae culture. *Artemia franciscana* metanauplii is widely used in cultured marine species (*Turcihan et al., 2021*). However, live foods such as *Artemia*, and rotifer contain a limited dietary value and might not provide all the necessary nutrients to develop and reproduce all species cultures (*Ostrowski & Laidley, 2001*). In contrast, copepods are often seen as live

prey in marine, freshwater, and brackish water ecosystems, constituting a substantial portion, up to 80%, of the zooplankton biomass in their respective native habitats (*Kimmerer et al., 2018*; *Yuslan et al., 2022a*). According to *Yuslan et al. (2022a)*, it has been shown that the cyclopoid copepod *Oithona rigida* exhibits a much higher concentration of highly unsaturated fatty acids (HUFAs) in comparison to *Artemia* and rotifer. However, the availability of copepods is limited due to their seasonal nature as zooplankton, which restricts their use primarily to hatcheries (*Yuslan et al., 2022b*). According to *Awal, Christie & Nieuwesteeg (2016)*, they have successfully cultivated amphipods using artificial substrates and confirmed that amphipods are higher in population, growth, and survival than natural substrates. However, there is still a lack of information and reports regarding farmers or aquaculture practices engaged in large-scale cultivation of amphipods. Besides that, *Drillet, Hansen & Kiørboe (2011)* have reported that the large-scale mysid culture has not been developed but is currently a very active area of research and development. This review is important to provide the available information on amphipods and mysids, including nutritional value, reproductive biology, culture method, and feed enhancement for amphipods and mysids. Aquaculture could make good use of amphipods and mysids as an alternative food source during the early culture of cephalopod, lobster, and shrimp species, and also as a live feed for seahorses, octopuses, and cuttlefish (*Vargas-Abúndez et al., 2021*). They are significant low-trophic position organisms that aid in the transfer of nutrients from the ocean to the coastline, a significant food source for economically significant fish species, and are crucial in the biological processing of algae inputs (*Jiménez-Prada, Hachero-Cruzado & Guerra-García, 2015*; *Lee et al., 2020*; *Jiménez-Prada, Hachero-Cruzado & Guerra-García, 2021*).

## SURVEY METHODOLOGY

Google Scholar, Web of Science, Springer Link, ScienceDirect, and Scopus resources were used to search articles from December 1965 to April 2023 that dealt with the use of *Artemia*, rotifer, copepods, cladocera, amphipods, and mysids in aquaculture, enrichment diets for live feeds, reproduction biology, and the culture technique of amphipods and mysids as a potential live feed in aquaculture. This review has included recent studies on marine and freshwater live feeds such as copepods and cladocera as guidance to emphasize the cultivation of new potential live feeds such as amphipods and mysids as marine fish food enhancement in hatcheries. This review included references to and excerpts from research and review articles written in English. We chose 142 articles in total to include in this review. The review did not include editorials, letters to the editor, or case studies.

### Morphology characteristics and distribution: Amphipods and Mysids
#### Amphipods

Amphipoda (Crustacea, Malacostraca) is a major aquatic, estuarine, and terrestrial freshwater taxon. They live in pelagic and benthic compartments with various life patterns, ecosystems, environmental requirements, and ecological feeding (*Podlesińska & Dąbrowska, 2019*). The previous study indicates that amphipods are commonly used for assessing the quality of marine and estuary sediment, mostly due to their habitat

preferences and lifestyle. Amphipods are herbivores, detritivores, scavengers, omnivores, or parasites and are important live feeds in marine ecosystems (*Podlesińska & Dąbrowska, 2019*). Amphipods such as gammarids and corophiids have laterally and dorsoventrally compressed bodies. According to *Hyne (2011)*, sexual dimorphism and reproductive strategies vary within species. By looking at the presence of marsupium, a mature female can be identified. In contrast, the mature male can be identified by the emergence of genital papillae on the body's ventral side (*Podlesińska & Dąbrowska, 2019*). In preparation for the next spawning, gravid females with embryos developing in the outer marsupial sac, oocyte maturation within their ovaries. In exchange, amphipods bear their broods, which extend their breeding cycle after spawning (*Hyne, 2011*). Amphipods are commonly used as model organisms for determining the quality of marine and estuarine sediments due to their nature and lifestyle (*Chapman, Wang & Caeiro, 2013*; *Podlesińska & Dąbrowska, 2019*). Epibenthic amphipods, such as gammarids, are abundant and ecologically important parts of marine benthic habitats since they interact closely with sediment and are easy to manage and culture in the laboratory (*Hyne, 2011*). As for Malaysian waters, the knowledge of amphipods has been poorly studied (*Lim, Azman & Othman, 2010*). Even though eleven taxa, including the new species of grammarian amphipods, have been recorded by *Azman & Othman (2013)* in Pulau Tioman waters, this results in additional documented amphipods and provides new information on the range and distribution pattern of amphipods in the South China Sea. The amphipod genus *Grandidierella* is comprised of more than 43 species that have been documented worldwide (*Wongkamhaeng et al., 2020*; *Shahin et al., 2023a*). The genus has a wide distribution in marine habitats, including brackish, estuarine, and coastal waters (*Azman & Othman, 2012*; *Myers & Desiderato, 2019*; *Myers, Sreepada & Sanaye, 2019*). In addition to their potential application as live food in aquaculture (*Jourde et al., 2013*; *Lo Brutto et al., 2016*), certain species are utilized as an indicator in studies on the toxicity of sediments (*Hindarti et al., 2015*). *G. halophila* is a newly discovered species from the Aoridae family, identified by *Wongkamhaeng, Pholpunthin & Azman (2012)*. It was found in the hypersaline waters of salt flats in the Samut Sakorn district, Thailand, located in the Inner Gulf of Thailand. The amphipods *G. halophila*, which are naturally abundant in the lagoons at Pantai Sri Tujuh in Kelantan, Malaysia, were discovered by *Shahin et al. (2023a)*. This discovery makes the amphipods easily accessible to researchers for their study as a potential new feed resource for aquaculture. Figure 1 shows a morphological diagram of adult amphipods.

### Mysids

The shrimplike crustaceans, Mysids that commonly referred to as "opossum shrimps," are common in both marine and freshwater environment (*Ogonowski et al., 2013*). Most mysids range in size from 5 to 25 mm, although deep-water species are generally larger, with the largest species known having lengths of up to 350 mm (*Porter, 2016*). Distinguishing mysid features include the presence of a statocyst in the uropod (but absent in six out of nine families) and, in females, the presence of a marsupium, also known as the brood pouch which is the basis of the group's common name, "opossum shrimps"

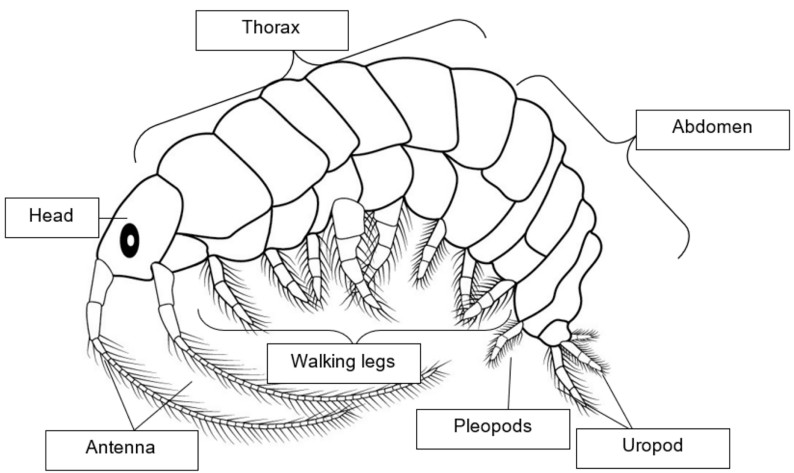

**Figure 1 Morphological diagram of adult Amphipods.**

(*Porter, 2016*). Mysid species are omnivorous and cannibalistic (*Berezina, Razinkovas-Baziukas & Tiunov, 2017*). The stomach of mysids gathered along the coast includes detritus, tiny crustacean bodies and appendages, and limited numbers of diatom shells (*Herrera et al., 2011*). Male mysids can be recognized by the presence of panels and secondary sexual characteristics, while females have a marsupium presence (*Ramarn, Chong & Hanamura, 2012*). Mysid shrimps are prevalent crustaceans found in a variety of aquatic environments such as oceans (~95%), estuaries, brackish water ecosystems, and freshwater lakes (*Viherluoto & Viitasalo, 2001*; *Miyashita & Calliari, 2016*). They are highly adaptable species and thus proficient in colonizing new territories (*Ketelaars et al., 1999*; *Viherluoto & Viitasalo, 2001*; *Rastorgueff et al., 2015*). Mysids are one of the most morphologically diverse classes of crustaceans (*Tan & Rahim, 2018*). They are good candidates for determining endocrine disruption due to the abundance of information available on their endocrinology (*Verslycke et al., 2004*). Mysids are highly abundant and widely distributed crustaceans, inhabiting various aquatic areas, with a particular preference for marine environments (*Gan et al., 2010*). Mysid is a significant constituent in the shallow coastal and estuarine waters, serving a crucial function in the transfer of energy from lower to higher trophic levels (*Mauchline, 1980*; *Mees & Jones, 1997*; *Yolanda et al., 2023*). The spatial distribution of crustacean zooplankton has been documented to be affected by tidal waves or water currents (*Hall & Burns, 2003*; *Macías et al., 2010*; *Yolanda et al., 2023*). There are 41 species of mysids recorded in Peninsular Malaysian waters (*Tan, Azman & Othman, 2014*), with the most common being *Erythrops minute*, *Mesopodopsis orientalis*, *Acanthomysis longispina*, *Acanthomysis quadrispinosa*, *Lycomysis spinicauda*, *Pseudanchialina inermis*, and *Prionomysis aspera*. The first mysid species from Malaysian waters was recorded by *Tattersall (1965)*, in the northern region of the Malacca Straits. Several mysid species belonging to the Anisomysini tribe have been found in the waters of Southeast Asia (*Sawamoto, 2014*; *Nurshazwan, Sawamoto & bin Abdul Rahim, 2021*). Two species, which is *Anisomysis* (*Anisomysis*) *aikawai* Ii, 1964 and *A.* (*Paranisomysis*)

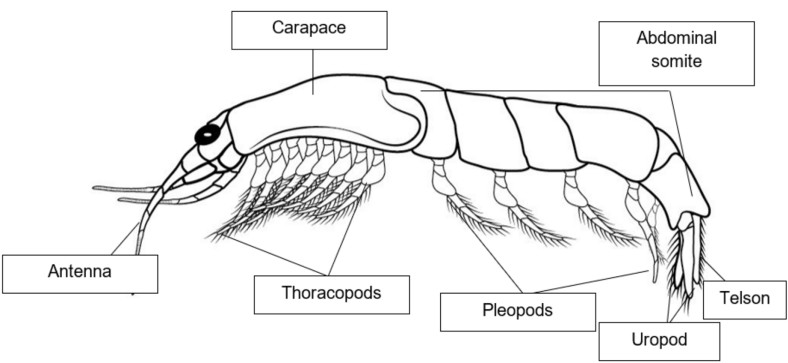

**Figure 2 Morphological diagram of adult Mysids.**

*ohtsukai* Murano, 1994, have been recorded from Malaysian waters (*Gan et al., 2010*; *Tan, Azman & Othman, 2014*; *Tan & Rahim, 2018*) and there was no record of any mysid of the genus *Idiomysis*. However, recent study by *Nurshazwan, Sawamoto & bin Abdul Rahim (2021)* has found another species of *Idiomysis*, which is *Idiomysis bumbumiensis* sp. nov. that was found at Pulau Bum Bum, Sabah, Malaysia. Species found is the seventh member of the genus *Idiomysis* and it is the first described in Southeast Asia (*Nurshazwan, Sawamoto & bin Abdul Rahim, 2021*). Figure 2 shows a morphological diagram of adult mysids.

## Reproductive biology of Amphipods and Mysids
### Amphipods
Amphipods have a direct development from juvenile to adulthood; after the juveniles are released from the females' brood chambers, they undergo several molts without metamorphosing before becoming adults (*Väinölä et al., 2008*; *Baeza-Rojano et al., 2010*). The molt cycle can be divided into four periods, which are post-molt, inter-molt, pre-molt, and ecdysis. The first stage, which starts at the end of the aged exoskeleton shedding, is known as an early post-molt period. The second stage starts with the incremental coloring of the antennas and legs. During the late post-molting cycle, the calcareous concrete within the posterior caeca is then used to mineralize the fresh skeleton. After that, during intermolting process, secretion of calcium carbonate and calcification of the last layer will continue to consolidate, which make up to 40% of the crustaceans' cuticle. When the new skeleton is gradually developed under the aged one, it shows that it is entering the pre-molting period (*Hyne, 2011*). Molting and oogenic processes are closely related to malacostracan crustaceans, which have a high degree of fecundity and body development (*Subramoniam, 2000*). In preparation for the next spawning, sexually mature females with developing embryos in their external marsupial pouches undertake concurrent oocyte maturation within their ovaries. Also, amphipods bear their broods beyond spawning, extending their breeding period. The initiation of molting in these species is postponed until their juveniles are hatched and released (*Sainte-Marie, 1991*). The molting of the rigid exoskeleton happens at the same time as the ovarian cycle. This lets the newly fertilized oocytes move to the oviducts through the marsupium while the exoskeleton is still strong

enough to let them through (*Hyne, 2011*). According to *Sainte-Marie (1991)*, *Baeza-Rojano et al. (2011)*, and *Shahin et al. (2023a)*, amphipods such as *Grandidierella halophila*, *Pontoporeia affinis*, *Orchestia mediterranea*, *Caprella grandimana*, and *Gammarus palustris* have iteroparous, semiannual, multivoltine life histories that produce multiple broods in a brief lifespan. The characteristics of tropical amphipods can be defined by their maturation time, size and length of brood, fecundity rate, and juveniles' development (*Cunha, Moreira & Sorbe, 2000b*; *Grabowski, Bacela & Konopacka, 2007*; *Baeza-Rojano et al., 2013b*; *Xue et al., 2013*; *Shahin et al., 2023a*, *2023b*). According to *Wang et al. (2009)*, a female of *Grandidierella japonica* can produce 13 to 17 juveniles from four broods in their lifespan. Previous studies have documented that among different species of amphipods, the number of embryos per brood varies. As an example, in average, *Cymadusa filosa* produces 20 juveniles, *Parhyale hawaiensis* produces 13 juveniles, *Elasmopus pectenicrus* produces seven juveniles, *Elasmopus levis* produces 22 juveniles, and *Eogammarus possjeticus* produces from 48 to 16 juveniles in their lifespan (*Borowsky, 1986*; *Aravind et al., 2007*). Understanding the potential fecundity of amphipods depends on a variety of aspects of their reproductive biology (*Wang et al., 2009*). *Cymadusa vadose*, a marine amphipod, was studied and reported by *Shahin et al. (2023c)* with the aim of assessing its potential as a new potential live feed. The amphipods that were obtained from the offshore at Bidong Island, Terengganu, were cultivated in filtered water with a salinity of 30 ppt at 28 °C in transparent plastic containers with a capacity of 500 mm. The life history of *C. vadosa* was observed by studying juveniles, individuals from a single offspring with five replicates. This species had a life history pattern characterized by two reproductive cycles each year and several generations. The duration required for the female to reach maturity was recorded as 17.4 days, with a corresponding size of 4.10 mm at the moment of maturation. The average duration of the incubation period was found to be 7.4 days. The average longevity for males was found to be 81.2 days, whereas females had a longer average lifespan of 113.6 days. The average number of broods generated during the course of an individual's lifetime was found to be 4.2 days. The brood sizes exhibited a wide range, spanning from 3 to 40 youngsters, with an average of 24.5 individuals per brood. The female population had a reproductive output of 103.0 offspring during their whole lifespan. A highly significant positive connection, with a coefficient of determination (R2) of 0.97, was observed between the size of female individuals and the quantity of offspring they generated. The observed mean maximum length of males and females across their lives was 5.95 and 6.11 mm, respectively. The initial examination of the life cycle of *C. vadosa* provides a basis for further investigations into its viability as a substitute live feed in aquaculture. Figure 3 shows the general life cycle of amphipods that have been modified from *Birmingham et al. (2005)*.

### Mysids

Numerous species of tropical mysids have been found to reproduce continuously throughout the year (*Hanamura, Siow & Chee, 2008*; *Hanamura et al., 2009*; *Biju & Panampunnayil, 2010*; *Biju, Gireesh & Panampunnayil, 2010*; *Ramarn, Chong & Hanamura, 2012*). Mysid reproductive biology resembles that of marsupials, including an
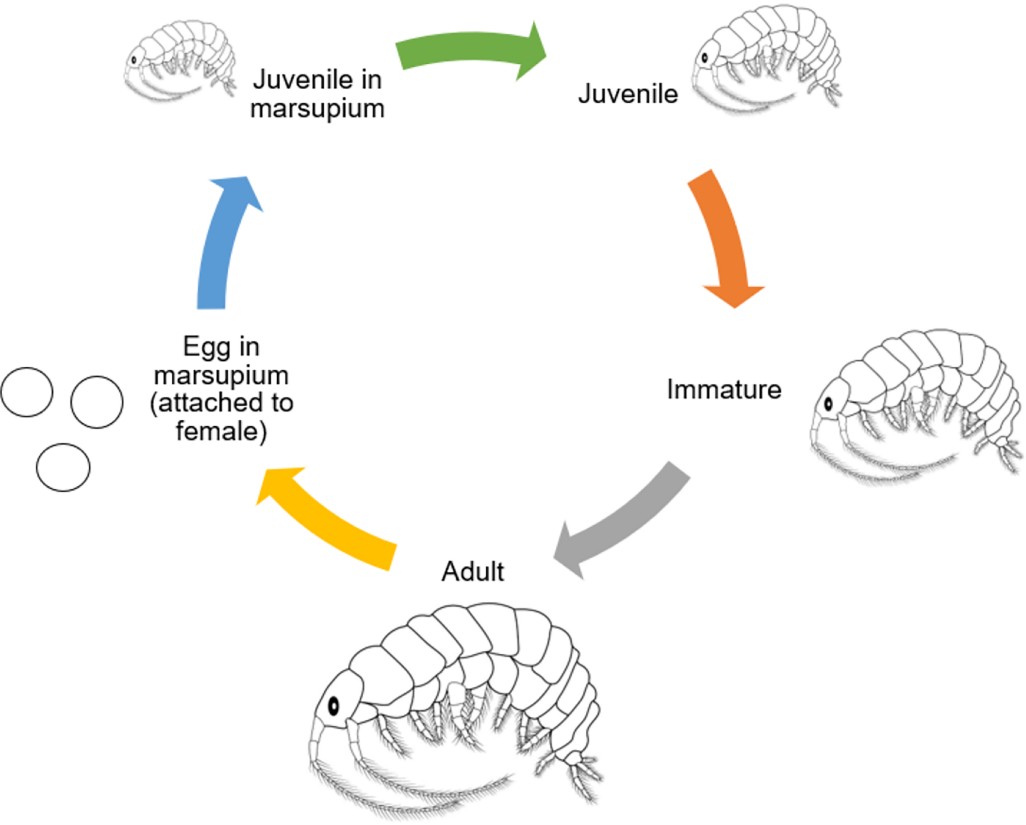

**Figure 3 General life cycle of Amphipods (modified from *Birmingham et al., 2005*).**

embryonic stage, a nauplioid stage, and a post-nauplioid stage (*Wittmann, 1981*; *San-Vicente, Guerao & Olesen, 2014*). Mysid female marsupium has three distinct but sequential stages of development, from oviposition to early embryonic stages (*Wortham-Neal & Price, 2002*). At some point during her ecdysis, the mother will release her young before laying another clutch of eggs in the marsupium. Oocytes undergo a secondary vitellogenic phase beginning on day two of the molten stage (*Verslycke et al., 2004*). In crustaceans, including amphipods, isopods, and decapods, secondary vellogenesis is cyclical and closely associated with the molten stage, serving as an example of a two-type form for the control of concurrent gonadal and somatic development (*Verslycke et al., 2004*). Therefore, the average number of mysids in females that are ovigerous could be used to guess how long the embryonic, nauplioid, and post-nauplioid phases last (*Mauchline, 1973*; *Wortham-Neal & Price, 2002*). The nauplioid stage, the embryonic stage, and the post-nauplioid stage occur in progressively shorter succession as development proceeds (*Mauchline, 1980*; *Delgado et al., 2013*). According to *Yolanda et al. (2023)*, mysids, *Rhopatophtalmus hastatus* brooding females with the nauplioid stage have a larger proportion than the brooding females with the embryonic and post-nauplioid stages. The mean body length among the brooding females, females with the naupiolid stage were the largest (10.73 ± 0.14 mm), followed by the post-naupiolid stage (10.67 ± 0.15) and embryonic phase (10.20 ± 0.13 mm) (*Yolanda et al., 2023*). While the number of offspring

in a given brood may vary greatly amongst mysid species, in general, tropical mysids are smaller overall and have fewer broods than their temperate counterparts, who are bigger overall and have more broods (*Mauchline, 1980*; *Fenton, 1992*; *Mees, Abdulkerim & Hamerlynck, 1994*). *Rhopalopthalmus hastatus* may lay as many as 17 eggs at once. *Rhopalopthalmus indicus*, *Rhopalopthalmus mediterraneus*, and *Rhopalopthalmus tattersallae* are all in the same genus and each have between 13 and 31 larvae (*Grabe, 1989*; *Baldó et al., 2001*; *Biju, Gireesh & Panampunnayil, 2010*). Brood loss or abortion during data collection, storage, and experimental processing accounts for differences in brood size, as stated by *Murtaugh (1989)* and *Paul et al. (2016)*. In contrast, the brood limits for temperate mysids like *Archaeomysis articulata*, *Gastrosaccus spinifer*, *Orientomysis japonica*, and *Neomysis integer* are 93, 162, 102, and 88, respectively (*Hanamura, 1999*; *Rappé et al., 2011*; *Akiyama, Ueno & Yamashita, 2015*). Little is known about the factors affecting brood size, although a study on *R. hastatus* by *Yolanda et al. (2023)* indicated a substantial correlation between the size of the brood and the length of the brooding females. This indicates that a larger marsupial mother is capable of producing and caring for a larger brood (*Saltzman, 1996*). Figure 4 shows the general life cycle of mysids that have been modified from *McKenney (2005)*.

## Culture method

### Amphipods

Despite the fact that amphipods are frequently found in their natural habitats in high numbers, it is necessary to develop a good culture and practice to produce a consistent supply of food for fish juveniles and ornamental marine species such as matured juveniles of seahorses (*Baeza-Rojano et al., 2013a*; *Vargas-Abúndez et al., 2021*). There is presently little information available on the use of amphipod culturing methods in aquaculture. A deeper knowledge of reproductive biology at the species level is essential to maximize mass production. The successful culture of amphipods on a small scale has been reported by *Baeza-Rojano et al. (2013a)*, *Soucek, Dickinson & Major (2016)*, and *Awal, Christie & Nieuwesteeg (2016)*, and intensive gammarid amphipod culture has only been performed under laboratory conditions for scientific research (*Delgado, Guerao & Ribera, 2011*); however, commercial-scale culture has not been reported. The culture method used by *Awal, Christie & Nieuwesteeg (2016)* uses a different type of substrate, which is seaweed and rope. Because of the habitat of amphipods, which can commonly be found in benthic habitats and live their whole lives by being bound to firm substrates (*Oliver et al., 2020*), these culture methods are vital for the reproduction of amphipods. As for the study by *Soucek, Dickinson & Major (2016)*, successful culturing methods for the amphipod *Hyalella azteca* on a small scale in the laboratory have been found by manipulating the amphipod's food. The use of different food combinations manages to provide details on the growth, survival, development, and reproduction of the amphipod *H. azteca*. Different culture methods for amphipods have been used by *Delgado, Guerao & Ribera (2011)*, *Baeza-Rojano et al. (2013b)*, *Awal, Christie & Nieuwesteeg (2016)*, *Vargas-Abúndez et al. (2021)*, and *Shahin et al. (2023b)*, depending on the amphipod's habitat and water parameters at the sampling area. As starter culture in the laboratory, the amphipods were

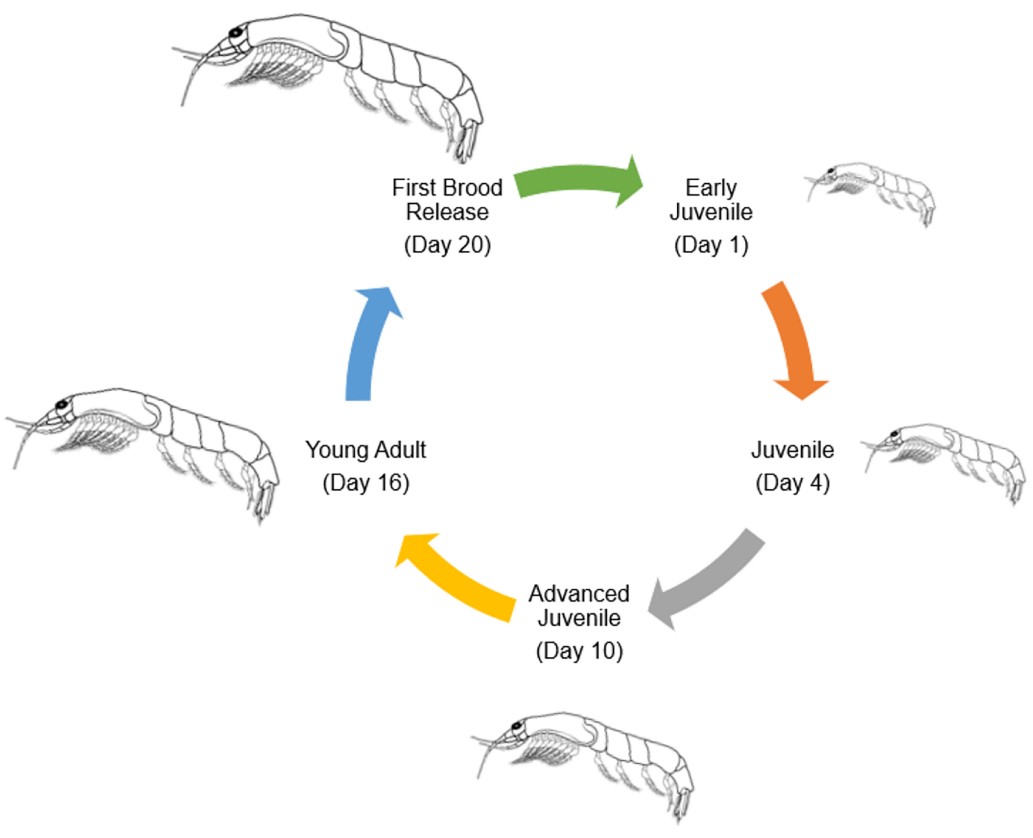

**Figure 4 General life cycle of Mysids (modified from *McKenney, 2005*).**

cultured in a 500 mL clear bowl or glass beaker (with strands of polypropylene rope as an artificial substrate) (*Awal, Christie & Nieuwesteeg, 2016*; *Shahin et al., 2023b*). After that, culture of amphipods will be upscaled to large cylindrical tanks, fiberglass tanks, or aquarium of 100 L until 171 L of water capacity (*Delgado, Guerao & Ribera, 2011*; *Baeza-Rojano et al., 2013a*; *Vargas-Abúndez et al., 2021*). During the culture period, amphipods were fed with mircoalgae, such as *Isochrysis galbana* and *Tetraselmis suecica* (*Baeza-Rojano et al., 2013a*), or commercial fish flake (*Shahin et al., 2023a*). Water parameters were maintained (mean ± standard deviation) at the range of 26.0 ± 0.5 °C–28 ± 1.0 °C, salinity at 7.0 ± 1.0 ppt–37.2 ± 0.5 ppt (range of salinity depends on the water quality in the sampling area where the amphipods were collected), and pH at the range of 8.1–8.3 (*Vargas-Abúndez et al., 2021*; *Shahin et al., 2023a*).

### Mysids

According to *Yolanda et al. (2023)*, in Malaysian waters, male adult mysids in estuaries mature earlier than females. While in the coastal area, females mature earlier than males. Temperate and tropical mysids differ in growth, maturity, and life duration; however, their productivity, performance, and maturation can still be clearly seen, and temperature is one of the key determinants that will influence their maturity. At 17 °C, male *Acanthomysis mitsukurii* developed sooner than females, while at 23 °C, females reached maturity earlier

**Table 1 Optimum water quality parameters for the culture of amphipods and mysids.**

| Parameters | Amphipods (*Caprella scaura*) | Mysids (*Neomysis awatschensis*) |
|---|---|---|
| **Temperature** | 18–23 °C | 20 °C |
| **Dissolved oxygen (DO)** | 5–9 ppm | 6.6–7.2 ppm |
| **Salinity** | 37–39 ppt | 30 ppt |

Note:
Optimum water quality parameters for the cultivation of potential amphipods (*Caprella scaura*) and mysids (*Neomysis awatschensis*) species that can be used in aquaculture hatcheries. Sources: *Baeza-Rojano et al. (2013a)*; *Lee et al. (2020)*.

than males (*Yamada & Yamashita, 2000*). A study by *Sudo, Kajihara & Noguchi (2011)* showed that *Orientomysis robusta* males matured in 13 to 32 days in the spring, 10 to 14 days in the summer, and 40 to 70 days in the fall and winter. While females were defined as mature in 15 to 35 days in the spring, 10 to 17 days in the summer, and 57 to 86 days in the fall and winter (*Sudo, Kajihara & Noguchi, 2011*). *Lee et al. (2020)*, also found that temperature is an important environmental factor for the mysid's marsupial development. Later, culturing mysids in a flow-through system has been studied. Through all the studies, they managed to get a lot of information on the method for culturing mysids (*Lee et al., 2020*). The previous study shows that the large-scale mysid culture has not been developed but is currently a very active area of research and development. According to *Lee et al. (2020)*, mysids, *Neomysis awatschensis*, were cultured in an automated aquaculture system with light:dark (L:D) photoperiod of 16:8 h, a temperature of 20 °C, a salinity of 30 ppt, a pH of 7.9–8.1, and a dissolved oxygen (DO) level of 6.6–7.2 mg L$^{-1}$. Apart from that, according to the latest study by *Griffin, O'Malley & Stockwell (2020)*, mysids were cultured in laboratory conditions by culturing them in a one-liter glass jar. Each glass jar was filled with 600 mL of chilled, dechlorinated water. *Mysis diluviana* was fed with three food treatment levels: *Daphnia* alone, detritus alone, and *Daphnia* plus detritus (*Daphnia* + detritus) (*Griffin, O'Malley & Stockwell, 2020*). Table 1 shows the water quality parameters of amphipod and mysid cultures that have been studied by *Baeza-Rojano et al. (2013a)* and *Lee et al. (2020)*. To achieve a successful culture, it is important to maintain an optimal range of water quality. Generally, poor water quality is related to excessive or harmful nitrogen component concentrations in water (*Rahman et al., 2023*).

## Nutritional value of Amphipods and Mysids

In order to increase the production of aquaculture, knowledge of the biochemical profile of marine organisms is vital for the discovery of new live feed that can be used effectively. According to *Leaver et al. (2008)*, fish larvae required essential fatty acids (EFAs) such as polyunsaturated fatty acids (PUFAs) and highly unsaturated fatty acids (HUFAs) for sustainable growth and survival. In addition, both amphipods and mysids also show adequate characteristics for use as live feed in aquaculture, such as suitable optimum sizes (0.3–2.5 cm), digestibility, and an adequate amount of lipids (10–15%) and protein (~40%) in dry weight (*Jiménez-Prada et al., 2018*; *Lee et al., 2020*; *Jiménez-Prada, Hachero-Cruzado & Guerra-García, 2021*).

**Table 2 Percentage of fatty acid composition in the gammaridea and caprellidea from the Strait of Gibraltar.**

| | Hyale perieri | Caprella penantis | Echinogammarus sp. | Caprella equilibra | Caprella grandimana | Elasmopus rapax | Jassa sp. | Caprella dilatata |
|---|---|---|---|---|---|---|---|---|
| **Saturated** | | | | | | | | |
| **16:0** | **24.83**[a] | **21.56**[b] | **16.98**[g] | **18.13**[e] | **17.59**[f] | **20.65**[c] | **19.71**[d] | **17.79**[f] |
| **17:0** | 0.54 | 1.44 | 1.38 | 1.20 | 1.16 | 1.09 | 1.37 | 0.93 |
| **18:0** | 3.65 | 5.20 | 4.46 | 4.56 | 3.53 | 5.27 | 6.32 | 4.43 |
| **Monounsaturated** | | | | | | | | |
| **18:1(n-9)** | **13.77**[c] | **12.57**[d] | **24.23**[a] | **11.44**[e] | **10.99**[f] | **17.79**[b] | **12.27**[d] | **10.84**[f] |
| **18:1(n-7)** | 5.24 | 2.56 | 3.64 | 3.00 | 7.23 | 2.39 | 2.45 | 1.96 |
| **18:1(n-5)** | 0.19 | 0.24 | 0.21 | 1.33 | 1.12 | 0.19 | 0.15 | 0.20 |
| **20:1(n-9)** | 1.10 | 1.47 | 1.22 | 0.52 | 0.52 | 1.16 | 1.18 | 1.65 |
| **Polyunsaturated** | | | | | | | | |
| **20:4(n-6)** | **5.43**[b] | **3.48**[d] | **2.36**[f] | **2.33**[f] | **10.25**[a] | **2.80**[e] | **2.14**[g] | **4.48**[c] |
| **20:5(n-3)** | **8.90**[f] | **15.87**[e] | **8.52**[g] | **0.32**[h] | **21.45**[a] | **16.10**[d] | **17.67**[b] | **17.14**[c] |
| **22:6(n-3)** | **2.08**[f] | **13.98**[b] | **0.86**[g] | **15.31**[a] | **7.72**[e] | **8.81**[d] | **11.84**[c] | **13.57**[b] |
| **DHA/EPA** | 0.23 | 0.88 | 0.10 | 0.69 | 0.28 | 0.55 | 0.67 | 0.79 |
| **DHA/AA** | 0.38 | 4.02 | 0.36 | 6.93 | 0.77 | 3.15 | 5.53 | 3.03 |
| **EPA/AA** | 1.64 | 4.56 | 3.61 | 10.10 | 2.73 | 5.75 | 8.26 | 3.82 |
| **Mon/PUFA** | 0.71 | 0.52 | 1.15 | 0.51 | 0.69 | 0.75 | 0.50 | 0.44 |
| **Mon/Sat** | 0.78 | 0.67 | 1.32 | 0.66 | 0.85 | 0.84 | 0.67 | 0.77 |

**Note:**
PUFA, Polyunsaturated; Mon, Monounsaturated; Sat, Saturated; AA, Arachidonic acid; EPA, Eicosapentaenoic acid; DHA, Docosahexaenoic acid. Source: *Baeza-Rojano, Hachero-Cruzado & Guerra-García (2014)*. The highest amount of each dietary component was highlighted as in bold. Lowercase letters (a–h) indicate significant different between each different species of gammaridae and caprellidea $p < 0.05$.

### Amphipods

*Baeza-Rojano, Hachero-Cruzado & Guerra-García (2014)* examined the lipids of various types of amphipods. The percentage of saturated fatty acids for each specimen ranges from 16.9% to 24%, monounsaturated fatty acids for each specimen ranges from 10.6% to 24.2%, and PUFAs for each specimen range from 8.51% to 17.7% for C20:5n-3, 0.8% to 13.9% for C22:6n-3, and 1.7% to 5.8% for C20:4n-6, according to *Baeza-Rojano, Hachero-Cruzado & Guerra-García (2014)* (Table 2). Through the study done by *Suontama et al. (2007)*, the nutritional value of Arctic amphipods *(Themsto libellula)* can support the growth and feed utilization of Atlantic halibut when meal from arctic amphipods is used to partially substitute fish meal in fish feed. According to *Moren et al. (2006)*, amphipods of the genus *Gammarus* have a high level of protein and beneficial PUFAs such as docosahexaenoic acid (DHA), eicosapentaenoic acid (EPA) (*Moren et al., 2006*) and they can provide the nutritional requirements for protein and lipids for marine larvae fish (*Kanazawa, 2003*). *Woods (2009)* stated that the amphipod caprellids contain moderately high amounts of beneficial PUFAs, including DHA (22:6n-3) and EPA (20:5n-3). Promising results have been obtained by previous studies when they explored amphipods as alternative protein and lipid sources in experimental diets for farmed fish (*Harlıoğlu & Farhadi, 2018*). Other than lipids, amino acid analysis has also been done in several studies. According to a study by *Fernandez-Gonzalez et al. (2018)*, arginine (23.8 mg/g), leucine (19.03 mg/g), and lysine

**Table 3 Comparison of fatty acid compositions (% of fatty acid methyl esters (FAME)) (Mean ± SD) of live prey organisms as food for developing fish and crustaceans' larvae.**

|  | Artemia[2] | Mysids | Copepods | Rotifer | Moina spp.[2] |
|---|---|---|---|---|---|
| **Total lipids** | ND[1] | 10.6 ± 0.1[a] | ND[1] | ND[1] | 9.84 ± 2.46[a] |
| **Fatty acids** |  |  |  |  |  |
| **C14:0** | 0.47 ± 0.05[d] | 3.1 ± 1.5[c] | 5.47 ± 0.09[a] | 0.98 ± 0.12[d] | 5.43 ± 0.05[a] |
| **C16:0** | 10.50 ± 0.25[e] | 26.5 ± 4.7[a] | 19.40 ± 0.15[c] | 17.0 ± 1.02[d] | 20.16 ± 0.25[b] |
| **C16:1n-7** | 1.46 ± 0.06[c] | 11.8 ± 2.5[a] | 4.61 ± 0.39[b] | 1.38 ± 0.34[c] | 3.38 ± 0.10[b] |
| **C18:0** | 6.57 ± 0.48[d] | 9.1 ± 0.6[b] | 4.55 ± 0.37[c] | 5.61 ± 0.43[c,d] | 13.24 ± 0.16[a] |
| **C18:1n-9** | 18.9 ± 0.31[a] | 7.8 ± 2.5[c] | 1.61 ± 0.39[d] | 7.92 ± 0.54[c] | 10.94 ± 0.04[b] |
| **C18:2n-6** | 5.29 ± 0.76[c] | 6.1 ± 1.4[c] | ND[1] | 22.2 ± 0.59[a] | 13.64 ± 0.07[b] |
| **C20:4n-6** | 0.48 ± 0.13[c] | 6.4 ± 1.3[a] | 0.44 ± 0.22[c] | 2.38 ± 0.56[b] | ND[1] |
| **C20:5n-3** | 2.19 ± 0.64[c] | 15.3 ± 1.6[a] | 6.61 ± 0.39[b] | 3.53 ± 0.31[c] | 0.66 ± 0.06[d] |
| **C22:6n-3** | 0.39 ± 0.03[d] | 13.2 ± 1.8[a] | 2.76 ± 0.21[c] | 5.05 ± 0.79[b] | ND[1] |
| **DHA: EPA** | 0.20 ± 0.07[c] | 0.9 ± 0.1[b] | ND[1] | 1.40 ± 0.29[a] | ND[1] |

**Notes:**
[1] ND: No data available.
[2] Source: *Das et al. (2007)*, *Eusebio, Coloso & Gapasin, 2010*, *Rocha et al. (2017)*, *Yuslan et al. (2022a)*.
Lowercase letters indicate significant different of fatty acid compositions between each different species of zooplankton, $p < 0.05$.

(18.8 mg/g) were the most prominent essential amino acids in amphipods. Glutamic (42.4 mg/g) and aspartic acids (29.8 mg/g) represented non-essential amino acids in amphipods. *Promthale et al. (2021)* examined the nutritional composition of different stages of dried amphipods, *Bemlos quadrimanu*s. The crude protein (CP) level of the juvenile stage (37.2 ± 1.0%) was significantly higher than that of the immature (29.6 ± 0.9%) and mature stages (25.9 ± 0.2%). The nutritional content of the *Gammarus pulex* study by *Abo-Taleb et al. (2020)* resulted in 40% of protein, 5.5% of fats, 27.4% of carbohydrates, and 2.9% of fiber. According to a study by *Jiménez-Prada et al. (2018)*, the five dominant species in this study had similar ash, protein, and carbohydrate compositions, but their total lipid compositions differed slightly (19.15% ± 0.48 and 18.35 ± 0.23, respectively, mean standard deviation).

### Mysids

According to Table 3, the total lipid percentage and fatty acid content of mysids is comparable with other live feeds used in industry, such as enriched copepods, rotifers, cladocera and *Artemia*. Based on Table 3, the total lipid content of mysids (10.6 ± 0.1) is higher than that of enriched *Artemia* (4.1 ± 1.1). The percentage of saturated fatty acids C16:0 in mysids (26% of FAME) are higher than in enriched *Artemia* (15.5% of FAME). The percentage of monounsaturated fatty acids C18:1n-9 of mysids (7.8% of FAME) lower than enriched *Artemia* (25.2% of FAME). A study by *Eusebio, Coloso & Gapasin (2010)* stated that, the percentage of PUFAs in mysids is higher than that of enriched *Artemia* (Table 3). *Eusebio, Coloso & Gapasin (2010)* stated that mysids contain a high nutritional value, such as protein, lipids, and beneficial fatty acids DHA, and EPA compared to enriched *Artemia*. Since live foods such as *Artemia*, rotifer, and copepod contain a limited

dietary value, they might not provide all the essential nutrients for the growth and survival of all species (*Ostrowski & Laidley, 2001*).

## Potential feed enhancement for Amphipods and Mysids

Research on marine amphipods and mysids has been stimulated by the need to find substitute live feed for aquaculture marine fish and crustaceans' species (*Shahin et al., 2023b*). Live food such as *Artemia*, rotifer, and copepod are common species that are used in marine aquaculture; however, to fill in the lack of long-chain HUFA, especially DHA in the rotifer and *Artemia*, they need to be enriched with a fatty acid (*Rasdi & Qin, 2016*; *Azani et al., 2023*). The same also goes for the amphipods and mysids, even though they already contain high nutritional value, as reported by *Woods (2009)* and *Eusebio, Coloso & Gapasin (2010)*, they still need to be fed with microalgae or zooplankton, which can help in increasing growth and population size. A nutritional enrichment of *Phronima pacifica*, a type of Amphipoda microcrustacean, has been studied by *Herawati et al. (2020)* by enriching them with two species of microalgae, which are *Chlorella vulgaris* and *Chaetoceros calcitrans*. Their study has found that *P. pacifica* mass-cultured with *C. vulgaris* resulted in the highest biomass, growth, and proteins and fats of *P. pacifica*. Gammaridan amphipod *Gammarus insensibilis* was proven by *Jiménez-Prada et al. (2018)* to possess all the characteristics for supplementing formulated diets in aquaculture due to its good biochemical composition, such as proteins, lipids, and amino acids for feed (live or dry), but they still need to be enriched with a fatty acid. The research by *Awal, Christie & Nieuwesteeg (2016)* on a different type of feed for caprellid amphipods reveals that the caprellids fed on *Phaeodactylum tricornutum* showed massive development and reproductive performance, resulting in an increase in population size. Other than microalgae and zooplankton as feed for amphipods, a study by *Jiménez-Prada, Hachero-Cruzado & Guerra-García (2021)* found that waste products such as detritus are useful for amphipod culture and provide a desirable biochemical profile. The use of detritus as a waste product to feed amphipods is an interesting topic due to its advantages of being cheap to produce.

Mysids are frequently used by aquarium hobbyists or in laboratory settings as a food source for a variety of aquatic species, such as cuttlefish, seahorses, and fish. They are also considered alternative live foods for the culture of marine species (*Oliveira et al., 2023*). In mysid culture, *Artemia* nauplii is usually used as feed; however, due to the expensive cost of *Artemia* cyst, a less expensive feed needs to be found to produce mass culture. Table 4 shows the major nutritional components and examples of live feed enrichment diets that were used by other researchers and farmers as guidelines to find a suitable diet for live feed culture and enrichment. Therefore, to meet the nutritional requirements of fish and prawn larvae, it is crucial to study the nutritional composition of enrichment diets so that it will contribute to the healthy development of fish during their early critical life stages. Live feeds that are enriched with essential nutrients are one of the crucial factors for the growth performance of fish larvae (*Ma & Qin, 2014*). Enrichment diets provide live feeds with the adequate amount of nutritional value necessary for survival, growth, and stability. Achieving natural live feeds, including amphipods and mysids, with higher

**Table 4 Example of feed used as an enrichment for live feed organisms and its nutritional compositions (%).**

|  | Protein | Lipid | Carbohydrate | Ash |
|---|---|---|---|---|
| *Chlorella vulgaris*[2] | $45.44 \pm 0.11^b$ | $10.47 \pm 0.12^c$ | ND[1] | $10.49 \pm 0.12^a$ |
| **Yeast** | $49.63 \pm 2.43^a$ | $4.64 \pm 0.52^d$ | $31.55 \pm 4.32^b$ | $7.98 \pm 0.76^c$ |
| **Rice bran** | $10.64 \pm 0.60^d$ | $21.84 \pm 0.54^a$ | $50.71 \pm 0.12^a$ | $10.08 \pm 0.12^b$ |
| **Palm kernel cake (PKC)** | $17.60 \pm 1.40^c$ | $5.50 \pm 0.30^d$ | $50.40 \pm 2.30^a$ | $6.10 \pm 1.20^c$ |
| **Soybean meal** | $19.00 \pm 0.10^c$ | $11.30 \pm 0.03^b$ | ND[1] | ND[1] |

Notes:
[1] ND, No data available.
[2] Sources: *Zarei et al. (2012)*, *Khan, Habib & Miah (2016)*, *Onofre et al. (2017)*, *Ilias et al. (2020)*, *Suhaimi et al. (2022b)*. Lowercase letters indicate significant different of nutritional compositions between each different feed use as an enrichment for live feed organisms, $p < 0.05$.

growth rates, stress tolerance, and good nutritional quality after enrichment are goals in the aquaculture industry in order to produce better-quality fish larvae with high growth rate (*Li, Zheng & Wu, 2021*). Numerous studies have focused on establishing methods for enhancing the nutritional quality of live feed with nutrients such as protein, lipids and fatty acids (*Kandathil Radhakrishnan et al., 2020*). In aquaculture, enriched zooplankton is essential for improving the nutritional value of fish and shellfish.

## Potential use of Amphipods and Mysids as alternative live feed in aquaculture

Amphipods and mysids have been recognized as crucial natural prey for various marine species. Studies on the nutritional effects of their partial substitution for fish meal in fish diets have produced promising results. Amphipods and mysids have been reported to be essential components of aquatic food webs in the wild because they act as conduits of nutrients and energy to higher trophic levels (*Shahin et al., 2023b*). Hence, their potential use for aquaculture species under captivity and laboratory conditions is ecologically acceptable. Live feeds, including amphipods and mysids, provide a suitable initial feed for fish and shrimp larvae compared to fish pellets due to their ability to swim in the water, which stimulates a feeding response in larvae, and also because they are high in digestibility and provide adequate nutrients (*Kandathil Radhakrishnan et al., 2020*). Protein and lipid are crucial for the growth, development, and survival of fish and shrimp larvae. A study on the use of amphipod meal as a fishmeal substitute on grey mullet has been conducted by *Ashour et al. (2021)*. Their result shows that amphipod meal with 50% partial replacement with fishmeal has benefited growth performance, feed utilization, and the histological and economic status of grey mullet (*Mugil cephalus*) fry. Table 5 shows a comparison of the proximate biochemical compositions of fishmeal and amphipod meal based on the study by *Ashour et al. (2021)*. According to their results, fishmeal has a higher amount of crude protein, while amphipod has produced a higher amount of ether extract and crude fiber than fishmeal.

The dependence on *Artemia* as a feed for newly hatched larvae greatly impacts the aquaculture industry. *Artemia* is essential to the marine fish and ornamental industries due to its high nutritional content to satisfy the need for fish culture. A recent study introduced

**Table 5  Proximate analysis of the fishmeal and amphipod meal (% of dry matter).**

|  | Fishmeal[1] | Amphipod meal[1] |
|---|---|---|
| Dry matter (%) | 89.39[b] | 93.15[a] |
| Crude protein (%) | 61.9 ± 1.8[a] | 25.9 ± 0.2[b] |
| Crude lipid (%) | 8.9 ± 1.1[a] | 4.7 ± 0.5[b] |
| Crude fiber (%) | 2.99 ± 2.90[b] | 11.2 ± 1.5[a] |
| Ash (%) | 22.4 ± 0.8[b] | 27.2 ± 0.5[a] |
| Nitrogen free extract | 10.76 ± 0.50[b] | 21.4 ± 0.6[a] |
| Gross energy (Kcal kg$^{-1}$) | 3,930.14[a] | 3,364.05[b] |

Notes:
[1] Sources: *Ween et al. (2017)*, *Kumar et al. (2018)*, *Bhuyain et al. (2019)*, *Ashour et al. (2021)*, *Promthale et al. (2021)*, *Alvanou et al. (2023)*.
Lowercase letters indicate significant different of nutritional compositions between fishmeal and amphipod meal, $p < 0.05$.

a new live feed such as amphipods and mysids to solve this problem. Research by *Baeza-Rojano et al. (2013b)* was the first to be able to grow caprellid amphipods in cultivation tanks big enough to achieve adequate densities for possible use as live feed for fish, with a maximum density of 10,460 individuals m$^{-2}$ and an average of 3,625 individuals m$^{-2}$ on the artificial substrates, which were dominated by juveniles. There has been a record of a few researchers using amphipods as feed for larvae rearing, such as in a study by *Vargas-Abúndez, Simões & Mascaró (2018)*, who have already experimented by feeding the amphipods to the lined seahorse *Hippocampus erectus*. Therefore, *Baeza-Rojano et al. (2010)* also reported that they have used the amphipods as alternative prey to culture cuttlefish *Sepia officinalis* hatchlings. Both of these experiments have shown the positive result that the amphipods can successfully help in the growth and development of lined seahorse *H. erectus* (*Vargas-Abúndez, Simões & Mascaró, 2018*) and hatchlings of cuttlefish *S. officinalis* (*Baeza-Rojano et al., 2010*). In addition, *González, Pérez-Schultheiss & López (2011)* also did a test on feeding the amphipod *Jassa marmorata* to the "Baby Octopus" *Robsonella fontaniana* paralarvae. Based on the research that has been done on amphipods, it shows that a suitable substrate and feed play an important role in successfully culturing amphipods. Different species of amphipods need a different type of substrate and feed according to their habitat. It shares the same concept as zooplankton such as copepods, where the availability of various food sources is essential to enhance the nutritional value of live feed culture in hatcheries (*Yuslan et al., 2022b*). Other than that, mysids also show great potential as an alternative live feed. Mysid is one of the major constituents in coastal areas and estuaries that is important in transporting nutrients in the form of energy at trophic levels from low to high (*Yolanda et al., 2023*). Mysids are also regarded as the primary source of food for fish and crustaceans (*Oh, Hartnoll & Nash, 2001*; *Link, Bolles & Milliken, 2002*; *Tomiyama, Uehara & Kurita, 2013*). Mysid culture has been recorded by *Wittmann (1984)*, *Domingues et al. (1998)*, and *Domingues et al. (1999)*.

Previous studies pointed out that the problem in the culture of mysids is the use of *Artemia* nauplii as a feed that is expensive and not economically sustainable for commercial culture. Aquaculturists and hatcheries are currently struggling with the cost of

producing cysts for commercial hatchery operations, as well as the labor and infrastructure costs associated with producing live *Artemia* to operate hatcheries. As a result, alternative zooplankton options are required for hatchery propagation (*Aziz et al., 2023*). Thus, to ensure the effectiveness of aquaculture activities, scientists are concentrating their efforts on the development of low-cost live feed alternatives (*Suhaimi et al., 2023*). Both amphipods and mysids show great potential as an alternative live feed in the aquaculture industry. However, research findings on these two live feeds are still limited. Furthermore, more research is needed to look into how to grow a lot of amphipods and mysids, how different water conditions affect the reproduction of amphipods and mysids, how to make their food better, and how to find species that are good for farming. These research efforts are crucial for gaining a comprehensive understanding of the potential future use of these live feeds in aquaculture.

## CONCLUSIONS

This review concludes by highlighting the possibility of raising amphipods and mysids for use as a potential live feed in aquaculture. Both amphipods and mysids have an adequate amount of protein, lipids, and essential PUFAs like DHA, EPA, and ARA, which makes them both very promising as live feeds that can provide the nutrients needed for fish and crustacean growth in their infancy. However, additional research on its nutritional compositions is necessary to develop a technique for mysid and amphipod mass culture. The aquaculture industry will be greatly impacted by the ability to mass produce amphipods and mysids because it will increase the amount of usable live feed available and possibly lower production costs in aquaculture hatcheries. Recent research has focused on the development of studies pertaining to amphipods and mysids globally. Its significant distribution has been seen globally, but future studies in other countries are recommended. Researchers believe that, *via* further investigation, amphipods and mysids exhibit promising potential as potential alternatives to *Artemia* in the future.

## ACKNOWLEDGEMENTS

The authors would like to thank Universiti Malaysia Terengganu (UMT), Malaysia, for the providence of facilities, lodging and other needs.

### Funding

This work was supported by the Malaysian Ministry of Higher Education (MOHE) under the (PRGS/1/2022/WAB04/UMT/03/1) Prototype Research Grant Scheme [Vot no. 54255]. The funders had no role in study design, data collection and analysis, decision to publish, or preparation of the manuscript.

## Grant Disclosures

The following grant information was disclosed by the authors:
Malaysian Ministry of Higher Education (MOHE): PRGS/1/2022/WAB04/UMT/03/1.
Prototype Research Grant Scheme: 54255.

## Competing Interests

The authors declare that they have no competing interests.

## Author Contributions

- Hidayu Suhaimi conceived and designed the experiments, performed the experiments, analyzed the data, prepared figures and/or tables, authored or reviewed drafts of the article, and approved the final draft.
- Muhammad Irfan Abdul Rahman conceived and designed the experiments, performed the experiments, analyzed the data, prepared figures and/or tables, authored or reviewed drafts of the article, and approved the final draft.
- Aisyah Ashaari conceived and designed the experiments, performed the experiments, analyzed the data, prepared figures and/or tables, authored or reviewed drafts of the article, and approved the final draft.
- Mhd Ikhwanuddin analyzed the data, authored or reviewed drafts of the article, and approved the final draft.
- Nadiah Wan Rasdi conceived and designed the experiments, performed the experiments, analyzed the data, prepared figures and/or tables, authored or reviewed drafts of the article, and approved the final draft.

## Data Availability

No raw data collected and analysed during finishing this literature review. Only articles from previous study were reviewed to highlights the important points of culturing amphipods and mysids as potential live feed in aquaculture.

## Supplemental Information

Supplemental information for this article can be found online at http://dx.doi.org/10.7717/peerj.17092#supplemental-information.

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

# PeerJ

**Herawati VE, Nailulmuna Z, Rismaningsih N, Hutabarat J, Pinandoyo P, Elfitasari T, Riyadi PH, Radjasa OK. 2020.** Growth performance and nutritional quality enrichment of *Phronima pacifica* by *Chlorella vulgaris* and *Chaetoceros calcitrans* as natural feed. *Biodiversitas Journal of Biological Diversity* **21(9)**:4253–4259 DOI 10.13057/biodiv/d210942.

**Herrera A, Gómez M, Molina L, Otero F, Packard T. 2011.** Rearing techniques and nutritional quality of two mysids from Gran Canaria (Spain). *Aquaculture Research* **42(5)**:677–683 DOI 10.1111/j.1365-2109.2010.02786.x.

**Hindarti D, Arifin Z, Prartono T, Riani E, Sanusi HS. 2015.** Toxicity of ammonia to benthic amphipod *Grandidierella bonnieroides*: potential as confounding factor in sediment bioasssy. *Indonesian Journal of Marine Sciences/Ilmu Kelautan* **20(4)**:215 DOI 10.14710/ik.ijms.20.4.215-222.

**Hyne RV. 2011.** Review of the reproductive biology of amphipods and their endocrine regulation: identification of mechanistic pathways for reproductive toxicants. *Environmental Toxicology and Chemistry* **30(12)**:2647–2657 DOI 10.1002/etc.673.

**Ilias NN, Rozalli NHM, Vy NHT, Eng HY. 2020.** Rice bran of different rice varieties in Malaysia: analysis of proximate compositions, antioxidative properties and fatty acid profile for data compilation. *Advances in Agricultural and Food Research Journal* **1(2)**:a0000164 DOI 10.36877/aafrj.a0000164.

**Jiménez-Prada P, Hachero-Cruzado I, Giráldez I, Fernández-Diaz C, Vilas C, Cañavate JP, Guerra-García JM. 2018.** Crustacean amphipods from marsh ponds: a nutritious feed resource with potential for application in Integrated Multi-Trophic Aquaculture. *PeerJ* **6**:e4194 DOI 10.7717/peerj.4194.

**Jiménez-Prada P, Hachero-Cruzado I, Guerra-García JM. 2015.** The importance of amphipods in diets of marine species with aquaculture interest of Andalusian coast. *Zool Baetica* **26**:3–29.

**Jiménez-Prada P, Hachero-Cruzado I, Guerra-García JM. 2021.** Aquaculture waste as food for amphipods: the case of *Gammarus insensibilis* in marsh ponds from southern Spain. *Aquaculture International* **29**:139–153 DOI 10.1007/s10499-020-00615-z.

**Jourde J, Sauriau PG, Guenneteau S, Caillot E. 2013.** First record of *Grandidierella japonica* Stephensen, 1938 (Amphipoda: Aoridae) from mainland Europe. *BioInvasions Records* **2(1)**:51–55 DOI 10.3391/bir.2013.2.1.09.

**Kanazawa A. 2003.** Nutrition of marine fish larvae. *Journal of Applied Aquaculture* **13(1–2)**:103–143 DOI 10.1300/J028v13n01_05.

**Kandathil Radhakrishnan D, AkbarAli I, Schmidt BV, John EM, Sivanpillai S, Thazhakot Vasunambesan S. 2020.** Improvement of nutritional quality of live feed for aquaculture: an overview. *Aquaculture Research* **51(1)**:1–17 DOI 10.1111/are.14357.

**Ketelaars HA, Lambregts-van de Clundert FE, Carpentier CJ, Wagenvoort AJ, Hoogenboezem W. 1999.** Ecological effects of the mass occurrence of the Ponto–Caspian invader, Hemimysis anomala GO Sars, 1907 (Crustacea: Mysidacea), in a freshwater storage reservoir in the Netherlands, with notes on its autecology and new records. *Hydrobiologia* **394**:233–248 DOI 10.1023/A:1003619631920.

**Khan AAI, Habib MAB, Miah MI. 2016.** Growth performance of *Chlorella vulgaris* in different concentrations of red sugar medium easily available in Bangladesh. *International Journal of Fisheries and Aquatic Studies* **6(6)**:136–141.

**Kimmerer WJ, Ignoffo TR, Kayfetz KR, Slaughter AM. 2018.** Effects of freshwater flow and phytoplankton biomass on growth, reproduction, and spatial subsidies of the estuarine copepod *Pseudodiaptomus forbesi*. *Hydrobiologia* **807**:113–130 DOI 10.1007/s10750-017-3385-y.

**Kumar M, Patel AB, Keer NR, Mandal SC, Biswas P, Das S. 2018.** Utilization of unconventional dietary energy source of local origin in aquaculture: impact of replacement of dietary corn with tapioca on physical properties of extruded fish feed. *Journal of Entomology and Zoology Studies* **6(2)**:2324–2329.

**Leaver MJ, Bautista JM, Björnsson BT, Jönsson E, Krey G, Tocher DR, Torstensen BE. 2008.** Towards fish lipid nutrigenomics: current state and prospects for fin-fish aquaculture. *Reviews in Fisheries Science* **16(sup1)**:73–94 DOI 10.1080/10641260802325278.

**Lee DH, Nam SE, Eom HJ, Rhee JS. 2020.** Analysis of effects of environmental fluctuations on the marine mysid *Neomysis awatschensis* and its development as an experimental model animal. *Journal of Sea Research* **156**:101834 DOI 10.1016/j.seares.2019.101834.

**Li X, Zheng S, Wu G. 2021.** Nutrition and functions of amino acids in fish. *Amino Acids in Nutrition and Health: Amino Acids in the Nutrition of Companion, Zoo and Farm Animals* **1285**:133–168 DOI 10.1007/978-3-030-54462-1_8.

**Lim JHC, Azman BAR, Othman BHR. 2010.** *Melitoid amhipods* of the genera Ceradocus Costa, 1853 and Victoriopisa Karaman and Barnard, 1979 (Crustacea: Amphipoda: Maeridae) from the South China Sea, Malaysia. *Zootaxa* **2348(1)**:23–39 DOI 10.11646/zootaxa.2348.1.2.

**Link JS, Bolles K, Milliken CG. 2002.** The feeding ecology of flatfish in the Northwest Atlantic. *Journal of Northwest Atlantic Fishery Science* **30**:1–17 DOI 10.2960/J.v30.a1.

**Lo Brutto S, Iaciofano D, Lubinevsky H, Galil BS. 2016.** *Grandidierella bonnieroides* Stephensen, 1948 (Amphipoda, Aoridae)—first record of an established population in the Mediterranean Sea. *Zootaxa* **4092(4)**:518–528 DOI 10.11646/zootaxa.4092.4.3.

**Ma Z, Qin JG. 2014.** Replacement of fresh algae with commercial formulas to enrich rotifers in larval rearing of yellowtail kingfish *Seriola lalandi* (Valenciennes, 1833). *Aquaculture Research* **45(6)**:949–960 DOI 10.1111/are.12037.

**Macías D, Somavilla R, González-Gordillo JI, Echevarría F. 2010.** Physical control of zooplankton distribution at the Strait of Gibraltar during an episode of internal wave generation. *Marine Ecology Progress Series* **408**:79–95 DOI 10.3354/meps08566.

**Mauchline J. 1973.** The broods of British mysidacea (Crustacea). *Journal of the Marine Biological Association of the United Kingdom* **53(4)**:801–817 DOI 10.1017/S0025315400022487.

**Mauchline J. 1980.** The biology of mysids and euphausiids. *Advances in Marine Biology* **18**:1–677 DOI 10.12691/marine-1-1-4.

**McKenney CL Jr. 2005.** The influence of insect juvenile hormone agonists on metamorphosis and reproduction in estuarine crustaceans. *Integrative and Comparative Biology* **45(1)**:97–105 DOI 10.1093/icb/45.1.97.

**Mees J, Abdulkerim Z, Hamerlynck O. 1994.** Life history, growth and production of *Neomysis integer* in the Westerschelde estuary (SW Netherlands). *Marine Ecology Progress Series* **109(1)**:43–57 DOI 10.3354/meps109043.

**Mees J, Jones MB. 1997.** Chapter 5. The hyperbenthos. In: Gibson RN, Barnes M, eds. *Oceanography and Marine Biology: An Annual Review.* Vol. 35. London: CRC Press, 221–256.

**Miyashita LK, Calliari D. 2016.** Distribution and salinity tolerance of marine mysids from a subtropical estuary, Brazil. *Marine Biology Research* **12(2)**:133–145 DOI 10.1080/17451000.2015.1099678.

**Moren M, Suontama J, Hemre GI, Karlsen Ø, Olsen RE, Mundheim H, Julshamn K. 2006.** Element concentrations in meals from krill and amphipods,—possible alternative protein sources in complete diets for farmed fish. *Aquaculture* **261(1)**:174–181 DOI 10.1016/j.aquaculture.2006.06.022.

**Murtaugh PA. 1989.** Fecundity of *Neomysis mercedis* holmes in lake Washington (Mysidacea). *Crustaceana* **57**:194–200 DOI 10.1163/156854089X00518.

**Myers AA, Desiderato A. 2019.** A new genus and species of Aoridae (Amphipoda, Senticaudata), *Propejanice lagamarensis* gen. nov. sp. nov. from Brazil. *Zootaxa* **4629(2)**:zootaxa-4629 DOI 10.11646/zootaxa.4629.2.11.

**Myers AA, Sreepada RA, Sanaye SV. 2019.** A new species of *Grandidierella coutière*, 1904, *G. nioensis* sp. nov. (Amphipoda, Aoridae), from the east coast of India. *Zootaxa* **4544**:119–124 DOI 10.11646/zootaxa.4544.1.7.

**Nurshazwan J, Sawamoto S, bin Abdul Rahim A. 2021.** *Idiomysis bumbumiensis* sp. nov., a new mysid species (Mysida, Mysidae, Anisomysini) from Southeast Asia. *Zoosystematics and Evolution* **97(2)**:345–354 DOI 10.3897/zse.97.68486.

**O'Malley BP, Bunnell DB. 2014.** Diet of *Mysis diluviana* reveals seasonal patterns of omnivory and consumption of invasive species in offshore Lake Michigan. *Journal of Plankton Research* **36(4)**:989–1002 DOI 10.1093/plankt/fbu038.

**Ogonowski M, Duberg J, Hansson S, Gorokhova E. 2013.** Behavioral, ecological and genetic differentiation in an open environment—a study of a mysid population in the Baltic Sea. *PLOS ONE* **8(3)**:e57210 DOI 10.1371/journal.pone.0057210.

**Oh CW, Hartnoll RG, Nash RD. 2001.** Feeding ecology of the common shrimp *Crangon crangon* in Port Erin Bay, Isle of Man, Irish Sea. *Marine Ecology Progress Series* **214**:211–223 DOI 10.3354/meps214211.

**Oliveira AF, Marques SC, Pereira JL, Azeiteiro UM. 2023.** A review of the order mysida in marine ecosystems: what we know what is yet to be known. *Marine Environmental Research* **188**:106019 DOI 10.1016/j.marenvres.2023.106019.

**Oliver JS, Hammerstrom KK, Kuhnz LA, Slattery PN, Oakden JM, Kim SL. 2020.** Benthic invertebrate communities in the continental margin sediments of the Monterey Bay area. In: Hendrickx ME, ed. *Deep-Sea Pycnogonids and Crustaceans of the Americas*. Cham: Springer, 193–235.

**Onofre SB, Bertoldo IC, Abatti D, Refosco D. 2017.** Chemical composition of the biomass of *Saccharomyces cerevisiae*-(Meyen ex EC Hansen, 1883) yeast obtained from the beer manufacturing process. *International Journal of Advanced Engineering Research and Science* **5(8)**:264258 DOI 10.22161/ijeab/2.2.2.

**Ostrowski AC, Laidley CW. 2001.** Application of marine food fish techniques in marine ornamental aquaculture: reproduction and larval first feeding. *Aquarium Sciences and Conservation* **3**:191–204 DOI 10.1023/A:1011349931035.

**Paul S, May DM, Lee M, Closs GP. 2016.** Body and brood sizes of *Tenagomysis* spp. (Crustacea: Mysida) in southern estuaries in New Zealand. *New Zealand Journal of Marine and Freshwater Research* **50(3)**:433–443 DOI 10.1080/00288330.2016.1154077.

**Podlesińska W, Dąbrowska H. 2019.** Amphipods in estuarine and marine quality assessment–a review. *Oceanologia* **61(2)**:179–196 DOI 10.1016/j.oceano.2018.09.002.

**Porter ML. 2016.** Collecting and processing mysids, stygiomysids, and lophogastrids. *Journal of Crustacean Biology* **36(4)**:592–595 DOI 10.1163/1937240X-00002443.

**Promthale P, Withyachumnarnkul B, Bossier P, Wongprasert K. 2021.** Nutritional value of the amphipod *Bemlos quadrimanus* sp. grown in shrimp biofloc ponds as influenced by different carbon sources. *Aquaculture* **533**:736128 DOI 10.1016/j.aquaculture.2020.736128.

**Punchihewa NN, Krishnarajah SR, Vinobaba P. 2017.** Mysid (Crustacea: Mysidacea) distribution in the Bolgoda estuarine system and Lunawa lagoon, Sri Lanka. *International Journal of Environment* **6(1)**:23–30 DOI 10.3126/ije.v6i1.16865.

**Rahman H, Azani N, Suhaimi H, Yatim SR, Yuslan A, Rasdi NW. 2023.** A review on different zooplankton culturing techniques and common problems associated with declining density. In: *IOP Conference Series: Earth and Environmental Science.* Vol. 1147, No. 1, Bristol, UK: IOP Publishing, 012012 DOI 10.1088/1755-1315/1147/1/012012.

**Ramarn T, Chong VC, Hanamura Y. 2012.** Population structure and reproduction of the mysid shrimp *Acanthomysis thailandica* (Crustacea: Mysidae) in a Tropical Mangrove Estuary, Malaysia. *Zoological Studies* **51(6)**:768–782.

**Rappé K, Fockedey N, Van Colen C, Cattrijsse A, Mees J, Vincx M. 2011.** Spatial distribution and general population characteristics of mysid shrimps in the Westerschelde estuary (SW Netherlands). *Estuarine, Coastal and Shelf Science* **91(2)**:187–197 DOI 10.1016/j.ecss.2010.10.017.

**Rasdi NW, Qin JG. 2016.** Improvement of copepod nutritional quality as live food for aquaculture: a review. *Aquaculture Research* **47(1)**:1–20 DOI 10.1111/are.12471.

**Rastorgueff PA, Bellan-Santini D, Bianchi CN, Bussotti S, Chevaldonné P, Guidetti P, Harmelin JG, Montefalcone M, Morri C, Perez T, Ruitton S, Vacelet J, Personnic S. 2015.** An ecosystem-based approach to evaluate the ecological quality of Mediterranean undersea caves. *Ecological Indicators* **54**:137–152 DOI 10.1016/j.ecolind.2015.02.014.

**Rocha GS, Katan T, Parrish CC, Gamperl AK. 2017.** Effects of wild zooplankton versus enriched rotifers and *Artemia* on the biochemical composition of Atlantic cod (*Gadus morhua*) larvae. *Aquaculture* **479**:100–113 DOI 10.1016/j.aquaculture.2017.05.025.

**Ros M, Guerra-García JM. 2012.** On the occurrence of the tropical caprellid *Paracaprella pusilla* Mayer, 1890 (Crustacea: Amphipoda) in Europe. *Mediterranean Marine Science* **13(1)**:134–139 DOI 10.12681/mms.30.

**Sainte-Marie B. 1991.** A review of the reproductive bionomics of aquatic gammaridean amphipods: variation of life history traits with latitude, depth, salinity and superfamily. *Hydrobiologia* **223**:189–227 DOI 10.1007/BF00047641.

**Saltzman J. 1996.** Ecology and life history traits of the benthopelagic mysid *Boreomysis oparva* from the eastern tropical Pacific oxygen minimum zone. *Marine Ecology Progress Series* **139**:95–103 DOI 10.3354/meps139095.

**Samir M, Banik S. 2015.** Production and application of live food organisms for freshwater ornamental fish larviculture. *Advances in Bio Research* **6(1)**:159–167.

**San-Vicente C, Guerao G, Olesen J. 2014.** Lophogastrida and Mysida. In: Martin JW, Olesen J, Høeg JT, eds. *Atlas of Crustacean Larvae.* Baltimore: Johns Hopkins University Press, 199–205.

**Sawamoto S. 2014.** Current status of mysid taxonomy in Southeast Asia. *Marine Research in Indonesia* **39(1)**:1–14.

**Shahin S, Okomoda VT, Ishak SD, Waiho K, Fazhan H, Azra MN, Azman AR, Wongkamhaeng K, Abualreesh MH, Rasdi NW, Ma H. 2023a.** Lagoon amphipods as a new feed resource for aquaculture: a life history assessment of *Grandidierella halophila. Journal of Sea Research* **192**:102360 DOI 10.1016/j.seares.2023.102360.

**Shahin S, Okomoda VT, Ishak SD, Waiho K, Fazhan H, Azra MN, Azman BAR, Wongkamhaeng K, Abualreesh MH, Rasdi NW, Ma H. 2023b.** First report on the life history of the marine amphipod *Ceradocus mizani* and its implication for aquaculture. *Invertebrate Biology* **142(2)**:e12398 DOI 10.1111/ivb.12398.

**Shahin S, Okomoda VT, Ishak SD, Waiho K, Fazhan H, Azra MN, Azman BAR, Wongkamhaeng K, Abualreesh MH, Rasdi NW, Ma H, Ikhwanuddin M. 2023c.** Life history traits of the marine amphipod *Cymadusa vadosa* under laboratory conditions: insights on

productivity and aquaculture potential. *Aquatic Sciences* **85(4)**:103
DOI 10.1007/s00027-023-01000-7.

**Soucek DJ, Dickinson A, Major KM. 2016.** Selection of food combinations to optimize survival,
growth, and reproduction of the amphipod *Hyalella azteca* in static-renewal, water-only
laboratory exposures. *Environmental Toxicology and Chemistry* **35(10)**:2407–2415
DOI 10.1002/etc.3387.

**Subramoniam T. 2000.** Crustacean ecdysterioids in reproduction and embryogenesis. *Comparative
Biochemistry and Physiology Part C: Pharmacology, Toxicology and Endocrinology*
**125(2)**:135–156 DOI 10.1016/s0742-8413(99)00098-5.

**Sudo H, Kajihara N, Noguchi M. 2011.** Life history and production of the mysid *Orientomysis
robusta*: high P/B ratio in a shallow warm-temperate habitat of the Sea of Japan. *Marine Biology*
**158**:1537–1549 DOI 10.1007/s00227-011-1669-8.

**Suhaimi H, Choy JLK, Yuslan A, Nasir A, Reduan A, Aizam NAH, Mubarak A, Rasdi NW.
2023.** The importance of low-cost live feed culture technology to the marine shrimp industry
during COVID-19. *Universal Journal of Agricultural Research* **11(2)**:358–370
DOI 10.13189/ujar.2023.110213.

**Suhaimi H, Yuslan A, Azani N, Habib A, Liew HJ, Rasdi NW. 2022a.** Effect of dietary enhanced
*Moina macrocopa* (Straus, 1820) on the growth, survival and nutritional profiles of hybrid Nile
tilapia fry. *The Egyptian Journal of Aquatic Research* **48(1)**:67–73
DOI 10.1016/j.ejar.2021.08.004.

**Suhaimi H, Yuslan A, Ikhwanuddin M, Yusoff FM, Mazlan AG, Habib A, Mustafa Kamal AH,
Rasdi NW. 2022b.** Effect of diet on productivity and body composition of *Moina macrocopa*
(Straus, 1820) (Branchiopoda, Cladocera, Anomopoda). *Crustaceana* **95(1)**:1–28
DOI 10.1163/15685403-bja10160.

**Suontama J, Karlsen Ø, Moren M, Hemre GI, Melle W, Langmyhr E, Mundheim H, Ringø E,
Olsen RE. 2007.** Growth, feed conversion and chemical composition of Atlantic salmon (*Salmo
salar L.*) and Atlantic halibut (*Hippoglossus hippoglossus L.*) fed diets supplemented with krill or
amphipods. *Aquaculture Nutrition* **13(4)**:241–255 DOI 10.1111/j.1365-2095.2007.00466.x.

**Tan HS, Azman BAR, Othman BHR. 2014.** Taxonomic status of mysid shrimps (Crustacea) from
Peninsular Malaysia waters. *Malayan Nature Journal* **66(3&4)**:103–116.

**Tan HS, Rahim AA. 2018.** Diversity of coastal mysids from Pulau Tinggi, Sultan Iskandar Marine
Park, Malaysia. *Nauplius–The Journal of the Brazilian Crustacean Society* **26**:1–12
DOI 10.1590/2358-2936e2018037.

**Tattersall OS. 1965.** Report on a small collection of Mysidacea from the northern region of the
Malacca Strait. In: *Proceedings of the Zoological Society of London*. Vol. 147. Oxford, UK:
Blackwell Publishing Ltd, 75–98.

**Tomiyama T, Uehara S, Kurita Y. 2013.** Feeding relationships among fishes in shallow sandy areas
in relation to stocking of Japanese flounder. *Marine Ecology Progress Series* **479**:163–175
DOI 10.3354/meps10191.

**Turcihan G, Isinibilir M, Zeybek YG, Eryalçın KM. 2022.** Effect of different feeds on
reproduction performance, nutritional components and fatty acid composition of cladocera
water flea (*Daphnia magna*). *Aquaculture Research* **53(6)**:2420–2430 DOI 10.1111/are.15759.

**Turcihan G, Turgay E, Yardımcı RE, Eryalçın KM. 2021.** The effect of feeding with different
microalgae on survival, growth, and fatty acid composition of *Artemia franciscana* metanauplii
and on predominant bacterial species of the rearing water. *Aquaculture International*
**29(5)**:2223–2241 DOI 10.1007/s10499-021-00745-y.

**Väinölä R, Witt JDS, Grabowski M, Bradbury JH, Jazdzewski K, Sket B. 2008.** Global diversity of amphipods (Amphipoda; Crustacea) in freshwater. *Freshwater Animal Diversity Assessment* 241–255 DOI 10.1007/s10750-007-9020-6.

**Vargas-Abúndez JA, López-Vázquez HI, Mascaró M, Martínez-Moreno GL, Simões N. 2021.** Marine amphipods as a new live prey for ornamental aquaculture: exploring the potential of *Parhyale hawaiensis* and *Elasmopus pectenicrus*. *PeerJ* **9**:e10840 DOI 10.7717/peerj.10840.

**Vargas-Abúndez JA, Simões N, Mascaró M. 2018.** Feeding the lined seahorse *Hippocampus erectus* with frozen amphipods. *Aquaculture* **491**:82–85 DOI 10.1016/j.aquaculture.2018.02.043.

**Verslycke TA, Fockedey N, McKenney CL Jr, Roast SD, Jones MB, Mees J, Janssen CR. 2004.** Mysid crustaceans as potential test organisms for the evaluation of environmental endocrine disruption: a review. *Environmental Toxicology and Chemistry: An International Journal* **23(5)**:1219–1234 DOI 10.1897/03-332.

**Viherluoto M, Viitasalo M. 2001.** Temporal variability in functional responses and prey selectivity of the pelagic mysid, Mysis mixta, in natural prey assemblages. *Marine Biology* **38**:575–583 DOI 10.1007/s002270000478.

**Wang R, Guan D, Yan Q, Han M, Chen H, Yan J. 2009.** Life history of amphipod *Grandidierella japonica* cultured in laboratory. *Marine Environmental Science* **28(3)**:272–274.

**Ween O, Stangeland JK, Fylling TS, Aas GH. 2017.** Nutritional and functional properties of fishmeal produced from fresh by-products of cod (*Gadus morhua* L.) and saithe (*Pollachius virens*). *Heliyon* **3(7)**:e00343 DOI 10.1016/j.heliyon.2017.e00343.

**Wittmann KJ. 1981.** On the breeding biology and physiology of marsupial development in Mediterranean *Leptomysis* (Mysidacea: Crustacea) with special reference to the effects of temperature and egg size. *Journal of Experimental Marine Biology and Ecology* **53(2–3)**:261–279 DOI 10.1016/0022-0981(81)90025-3.

**Wittmann KJ. 1984.** Oceanography and Marine Biology. An Annual Review. In: Margaret B, ed. *With 5 Plates, 83 Figs*. Vol. 21. Aberdeen: Aberdeen University Press 1983, 591 DOI 10.1002/iroh.19840690513.

**Wongkamhaeng K, Dumrongrojwattana P, Shin MH, Boonyanusith C. 2020.** *Grandidierella gilesi* Chilton, 1921 (Amphipoda, Aoridae), first encounter of non-indigenous amphipod in the Lam Ta Khong River, Nakhon Ratchasima Province, North-eastern Thailand. *Biodiversity Data Journal* **8**:e46452 DOI 10.3897/BDJ.8.e46452.

**Wongkamhaeng K, Pholpunthin P, Azman BAR. 2012.** *Grandidierella Halophilus* a new species of the family Aoridae (Crustacea: Amphipoda) from the saltpans of The Inner Gulf of Thailand. *The Raffles Bulletin of Zoology* **60(2)**:433–447.

**Woods CM. 2009.** Caprellid amphipods: an overlooked marine finfish aquaculture resource? *Aquaculture* **289(3–4)**:199–211 DOI 10.1016/j.aquaculture.2009.01.018.

**Wortham-Neal JL, Price WW. 2002.** Marsupial developmental stages in *Americamysis bahia* (Mysida: Mysidae). *Journal of Crustacean Biology* **22(1)**:98–112 DOI 10.1163/20021975-99990213.

**Xue S, Fang J, Zhang J, Jiang Z, Mao Y, Zhao F. 2013.** Effects of temperature and salinity on the development of the amphipod crustacean *Eogammarus sinensis*. *Chinese Journal of Oceanology and Limnology* **31(5)**:1010–1017 DOI 10.1007/s00343-013-2302-0.

**Yamada H, Yamashita Y. 2000.** Effects of temperature on intermolt period, growth rate and reproduction rate in *Acanthomysis mitsukurii* (Crustacea: Mysidacea). *Crustacean Research* **29**:160–169 DOI 10.18353/crustacea.29.0_160.

**Yolanda R, Ambarwati R, Rahayu DA, Rahim AA, Sriwoon R, Lheknim V. 2023.** Population structure and reproductive biology of the mysid *Rhopalophthalmus hastatus* Hanamura,
Murano & Man, 2011 (Crustacea: Mysida) in the Songkhla Lagoon System, southern Thailand. *Thalassas: An International Journal of Marine Sciences* **39(1)**:287–300 DOI 10.1007/s41208-022-00490-w.

**Yuslan AA, Ghaffar M, Arshad A, Suhaimi H, Ikhwanuddin M, Rasdi NW. 2022b.** Enhancement of protein, lipid and fatty acids in copepod, *Oithona rigida* for the improvement of growth, development and survival of early-stage mud crab larvae (*Scylla olivacea*). *Aquaculture Research* **53(17)**:6158–6171 DOI 10.1111/are.16089.

**Yuslan A, Suhaimi H, Taufek HM, Rasdi NW. 2022a.** Effect of bio-organic fertilizer and agro-industrial residue on the growth and reproduction of cyclopoid copepod, *Oithona rigida* (Giesbrecht, 1896). *International Journal of Aquatic Biology* **10(2)**:151–168.

**Zarei M, Ebrahimpour A, Abdul-Hamid A, Anwar F, Saari N. 2012.** Production of defatted palm kernel cake protein hydrolysate as a valuable source of natural antioxidants. *International Journal of Molecular Sciences* **13(7)**:8097–8111 DOI 10.3390/ijms13078097.