# Peer review of "Adaptation and potential culture of wild Amphipods and Mysids as potential live feed in aquaculture: a review"

_PeerJ, doi:10.7717/peerj.17092_

## Round 0.1 · original submission · Major Revisions

Dear Authors

The reviewers have commented on your manuscript. You can find them in the attached reports. Reviewers highlighted outnumbered critical errors in the manuscript; therefore, a major revision is needed for your article.

I request that you check and correct the manuscript based on the reports.

**Language Note:** The review process has identified that the English language must be improved. PeerJ can provide language editing services - please contact us at copyediting@peerj.com for pricing (be sure to provide your manuscript number and title). Alternatively, you should make your own arrangements to improve the language quality and provide details in your response letter. – PeerJ Staff

·

Basic reporting

Line 41 – Larvae and fingerling of ????
Line 44 – latin name of rotifer should be added
Line 69 – I do not agree this statement. Copepods are rich in essential nutrients and they are natural feeds of many fish. Rotifer and Artemia are lack of essential nutrients and that’s why, they are enriched.
Line 71- explain in paranthesis HUFA (Highly unsaturated fatty acids)

Line 66 – 74 – It is not enough that knowledge to sum up importance of common live preys (rotifer, artemia and copepods). I mean if this paper is review, past, current and future of those live preys should be well explained. So, then pls, expand this paragraph before dig in to amphipods and mysids culture. I advice couple of references below:

• Turcihan, G., Turgay, E., Yardımcı, R. E., & Eryalçın, K. M. (2021). The effect of feeding with different microalgae on survival, growth, and fatty acid composition of Artemia franciscana metanauplii and on predominant bacterial species of the rearing water. Aquaculture International, 29(5), 2223-2241.

• Turcihan, G., Isinibilir, M., Zeybek, Y. G., & Eryalçın, K. M. (2022). Effect of different feeds on reproduction performance, nutritional components and fatty acid composition of cladocer water flea (Daphnia magna). Aquaculture Research, 53(6), 2420-2430.

Line 93-94 / Author says alternative live feeds in aquaculture were checked by Google scholar, Sciencedirect, WOS, Springer Link. In fact, I believed that author did not refer recent studies. If the live feeds are subject, other live preys also should be mentioned. For instance, cladocer culture is very important and recent studies have been reported for direct utilization and/or feed ingredients.

Line 233 – remove one of ‘’in’’
Line 234 – put space after comma

Culture methods should define their culture vessels and foods in details. And further offers should be represented. Overall, culture methods should be expanded by recent articles with figures.

Line 282 – Pollyunsaturated fatty acids, highly unsaturated fatty acids terms should be defined where first mentioned and consistency should be sustain throughout the manuscript.

Line 284 – pls, use ‘’digestibility’’ instead of ‘’digestible’’
Line 285 – those value in dry weight ?
Line 290 – is that ‘’%’’ and SD value ?
Line 292 – fatty acids should be C20:5n-3

Line 288 – this section should be riched by listed references below:

• Promthale, P., Withyachumnarnkul, B., Bossier, P., & Wongprasert, K. (2021). Nutritional value of the amphipod Bemlos quadrimanus sp. grown in shrimp biofloc ponds as influenced by different carbon sources. Aquaculture, 533, 736128.

• A Abo-Taleb, H., F Zeina, A., Ashour, M., M Mabrouk, M., E Sallam, A., & MM El-feky, M. (2020). Isolation and cultivation of the freshwater amphipod Gammarus pulex (Linnaeus, 1758), with an evaluation of its chemical and nutritional content. Egyptian Journal of Aquatic Biology and Fisheries, 24(2), 69-82.

• Jiménez-Prada, P., Hachero-Cruzado, I., Giráldez, I., Fernández-Diaz, C., Vilas, C., Cañavate, J. P., & Guerra-García, J. M. (2018). Crustacean amphipods from marsh ponds: a nutritious feed resource with potential for application in Integrated Multi-Trophic Aquaculture. PeerJ, 6, e4194.

• Fernandez-Gonzalez, V., Toledo-Guedes, K., Valero-Rodriguez, J. M., Agraso, M. D. M., & Sanchez-Jerez, P. (2018). Harvesting amphipods applying the integrated multitrophic aquaculture (IMTA) concept in off-shore areas. Aquaculture, 489, 62-69.


Figure 3 and 4 / should include small individiuals.

Table 2 and 3 should be improved. Pls, include other fatty acids.

At Table 3, how it is possible that enriched Artemia did not contain DHA ? something wrong in here.

Table 4 should include SD values

Table 5 – pls add SD values. Ether extract ? Is that Crude Lipid ?

Experimental design

no comment

Validity of the findings

no comment

Additional comments

no comment

Reviewer 2 ·

Basic reporting

Gratitude is expressed to the authors for conducting this study. In the study, the potential uses of Amphipod and Mysid species in aquaculture have been compiled. Unfortunately, A major revision of the article is recommended.
1. The article has been examined to assess its clarity, precision, and adherence to professional English standards. Specific sentences that could benefit from revision have been identified; examples have been provided below. It is recommended that the entire text be reviewed to enhance its overall clarity and professionalism.
Corrected versions:
Line 18-19: Live food such as phytoplankton and zooplankton are essential food sources in aquaculture.
Line 20-22: Artemia and rotifer are commonly live feed used in aquaculture; each feed has limited dietary value, which is unsuitable for all cultured species.
Line 86-88: They are also considered a significant food source for economically significant fish species (Jimenez-Prada, Hachero-Cruzado, & Guerra-Garca, 2015; Lee et al., 2020).
Line 117-119: Epibenthic amphipods, such as gammarids, are abundant and ecologically important parts of marine benthic habitats since they interact closely with sediment and are easy to manage and culture in the laboratory (Hyne, 2011).
Line 502-503: However, additional research on its nutritional compositions is necessary to develop a technique for mysid and amphipod mass culture.

Experimental design

2. In the introduction of the review article, relevant literature has been utilized. The information provided for the introduction and the references used are considered sufficient.
3. The article is by the Aims and Scope of the journal. The methods used in the review article have been described in detail. The number of articles reviewed has been specified.

Validity of the findings

4. The article has compiled literature on the morphologies, reproductive cycles, cultivation methods, nutritional contents, and alternative uses in aquaculture of Amphipods and Mysids. Therefore, the article is believed to meet the objectives in the introduction. However, it is thought that the improvement of the areas specified below by the authors would be significant in enhancing the quality of the article.
Recommendations:
Information has been limited to the Malaysia and China Sea in lines 119 – 124 for Amphipods. However, it might be more appropriate to add comments on studies conducted worldwide within the scope of the literature.
Information has been limited to the Malaysia and China Sea in lines 149-152 for Mysids. However, it might be more appropriate to add comments on studies conducted worldwide within the scope of the literature.
In the table description, the readers need to determine which data comes from which source that literature information is provided within the table. Therefore, it is kindly requested that the tables be reconstructed following my suggestion.
Table 2 and Table 5 have been created using only a single source. It would be more beneficial for the journal readers if the table were enhanced using various sources.

Additional comments

5. It is believed that the scientific value of the article would be enhanced if the Conclusion section were rewritten to encompass the literature mentioned above and include unresolved questions. Additionally, it would be better to mention future directions in the Conclusion section.

·

Basic reporting

There were many small instances of the English needing correcting. Here is a list of the corrections I made:

line 28: change to "that are required".

line 32: change to "research which has been".

line 34: change to "paper is intended".

line 36: change to "researchers".

line 52: change "either" to "whether".

line 54: change to "They are a dominant species of the benthic fauna and often have high diversity"

line 55-57: this sentence is confusing, please clarify.

line 103-104: please clarify this sentence.

line 120: change to "taxa".

line 127: I think "cm" is meant to be "mm".

line 157: change "has" to "have".

line 165: change to "continue"

line 169-171: the grammar in this sentence is not quite right, please correct.

line 189: please correct the citation.

line 225: maybe change to something like "Little is known about the factors affecting brood size".

line 233: delete repeated word.

line 249: change to "benthic habitats". On the next line, change "this" to "these".

line 251: change to "successful culturing methods for the amphipod"

line 257: maybe change "they" to "researchers" or something, to make this sentence more understandable.

line 271: change to "cultured marine species".

line 283: change to "show".

line 294: delete "to".

line 316: change to "compared".

line 331: change to "enriching"

line 337: delete "Through".

line 337: delete "are".

line 342: change "to be" to "is".

line 347: change to "a variety of"

line 349: change to "are usually". On the next line change to "need to be found".

line 352: change to "researchers".

line 355: I did not understand this sentence, please clarify.

line 373: change "than" to "compared to". On the next line, change to "and also because they are high in digestibility".

line 382:change to "a higher amount" for both instances.

line 394: change to "is able".

line 396: change to "was the first".

line 406: change to "results".

line 417: change to "increasing". On the next line delete "in".

line 422: change to "was observed when it was fed with"

line 431: change to "these experiments have". On the next line, I did not understand the use of the word "susceptibility", maybe delete.

line 433-434: Please clarify this sentence.

line 437: change to "culture".

line 454: change to "deterioration"

line 461: change to "influences".

line 464: delete "Although".

line 479: change to "a study on alternative diets for mysids has been done".

line 485-487: please clarify this sentence.

line 489: change to "ensure".

line 493-495: this sentence is difficult to understand, please clarify.

Experimental design

The citations are done well through most of the manuscript, but for the paragraph lines 257-275 there are many facts stated that do not have a citation and it is difficult to determine where the information came from, so in this whole paragraph, please make sure each fact stated is accompanied by a citation.

Validity of the findings

no comment

Additional comments

line 19: is it correct to say "benthic" fauna? Normally plankton are suspended in the water and so are pelagic.

line 102: I would change the word "species" to "taxon".

line 106: I am not sure the word "compounds" is right to use here, maybe change to something else.

Table 3: what are the errors shown after each of the numbers, e.g. standard error, standard deviation, 95% confidence interval, or something else? Also I guess these values are averages, so please state that averages are shown in the table caption.

---

## Round 0.2 · Minor Revisions

Dear Authors

The reviewers have commented on your manuscript. You can find them in the attached reports. Reviewers highlighted a few errors in the revised manuscript; therefore, a minor revision is needed for your article.

I request that you check and correct the manuscript based on the reports.

Sincerely yours

·

Basic reporting

Manucscript include important data on mphipods and Mysids. Those live preys can be used in several ways in aquaculture such as direct or feed ingredients. Therefore, topic is important.

Experimental design

It is review paper and it is improved after revision. However, I did not see that all tables include statistics differences with upper superscript letters. I only concern about it. If they added sig. superscripts in tables, it is fine.

Validity of the findings

The manuscript includes important data and become good review paper.

Additional comments

-

·

Basic reporting

The use of English language was done well, I only saw three minor corrections - line 131: change "have" to "has", line 243: change to "males", line 320: delete "at".

Experimental design

no comment

Validity of the findings

no comment

Additional comments

line 47: are they really commonly found in colonies? Mostly I think of the way amphipods live as being describable as individual.

line 69: a massive rate of what?

line 131-147: I found it difficult to understand the relevance of this section, given the review is focused on feed for aquaculture, e.g. why bring up about two amphipods recently found in lava field groundwater? (line 138) (have they been found to be useful for being cultured or something?). Is there some relevance of the species discovered in the alps? (line 142-143). Overall it seems like just an incomplete and not very relevant collection of facts, so the section might be deleted or made more relevant for the review. And this also applies to the text on line 167-197. These sections could be improved by having the many specific details more synthesized into take-home messages for the reader about amphipods and mysids in general.

line 273-274: as all species grow they have increasing body size through time.

line 387: because this is a review, I do not think its necessary to even mention differences from the reviewed papers that were non-significant (the whole point of a review is to pick out only the important and relevant facts from the literature).

Table 1: maybe state the species names of the amphipods and mysids in these two studies, so the reader knows the data is only relevant for those limited numbers of species (not to amphipods and mysids in general).

Figure 3-5: all these figures are only very basic with only limited information in them (Fig. 5 especially is unnecessary). The information in all the these figures could probably be explained more succinctly in just a few sentences of text - I would delete these figures.

---

## Round 0.3 · accepted · Accept

Dear Dr. Rasdi

I would like to thank you and your co-authors for making the corrections and changes requested by the reviewers. I read and checked carefully your valuable article and I am happy to inform you that your article has been accepted for publication in PeerJ.

Best regards